# Information provision and decision-making in the treatment of abdominal aortic aneurysm: A qualitative study of patient experience

**Jan Lecouturier**[1]*, **Gerry Stansby**[2,3], **Richard G. Thomson**[1]

**1** Population Health Sciences Institute, Newcastle University, Newcastle upon Tyne, United Kingdom, **2** Northern Vascular Centre, Newcastle upon Tyne Hospitals NHS Foundation Trust, Newcastle upon Tyne, United Kingdom, **3** School of Surgical and Reproductive Sciences, Newcastle University, Newcastle upon Tyne, United Kingdom

* jan.lecouturier@newcastle.ac.uk

## Abstract

### Introduction

Shared decision making (SDM) refers to patients and health care professionals working together to reach a decision about treatment/care. In abdominal aortic aneurysm (AAA) treatment options are influenced by patients' clinical characteristics, their preferences, and potential trade-offs between alternative interventions. This is a prime example of where SDM is essential to ensure the right decision is made for the right patient, yet we have little understanding of what happens in practice. This study explored patient experiences to understand SDM practice in AAA surgery.

### Methods

We used a qualitative approach to describe, and identify improvements to, current treatment decision making in abdominal aortic aneurysm (AAA) surgery. Two groups of patients were interviewed: those at the point of discussing treatment options (with corresponding digitally recorded consultation data) and following surgical intervention from one hospital. Framework analysis was used.

### Results

Fifteen patients were interviewed, seven at the point of discussing treatment options and eight following surgical intervention. Timing, format and sources of information, verbal framing of interventions and level of patient engagement were key themes. Four areas for improvement were identified: earlier provision and more detailed written information along with signposting to quality on-line information; both intervention options, risks, benefits, and consequences, were not always discussed; some clinicians were somewhat directive in the decision-making process; and patients' treatment values/preferences were not explored—the only example was in one of the eight recorded consultations. Patients could feel overwhelmed by the information and decision and fearful of the impending surgery.

National Research Ethics Service Committee West Midlands - South Birmingham. REC reference: 15/WM/0307. southbirmingham.rec@hra.nhs.uk.

**Funding:** GS and RT received the award. The study was funded by TargetPAD [grant number RES/0150/7857]. The funders had no role in study design, data collection and analysis, decision to publish, or preparation of the manuscript.

**Competing interests:** The authors have declared that no competing interests exist.

## Conclusions

More emphasis should be placed on the provision of full information and the exploration of patient values and preferences for treatment. Clinician training and support for patients, including decision aids, could facilitate the decision-making process. Providing written information earlier and guidance on reliable on-line resources would benefits patients and their families.

## Introduction

Aortic aneurysm is a swelling of the aorta, the main blood vessel that brings blood away from the heart. Management options include surgery (open and endovascular repair), surveillance or conservative treatment. Abdominal aortic aneurysm (AAA) is a disease primarily of men (4:1 male to female) [1]. Patients with AAA tend to be older and often have risk factors that influence the outcomes of surgery and/or mortality. Abdominal aortic aneurysm (AAA) surgery is an example of where options are influenced by patients' clinical characteristics, their preferences, and potential trade-offs between alternative interventions. Therefore, consideration of the most appropriate type of surgery, or whether the aneurysm is best managed conservatively, needs to be decided on a case-by-case basis. This is a prime example of where shared decision making (SDM) is essential to ensure the right decision is made for the right patient. SDM involves a person and their health care professional working together to reach a decision about treatment/care [2, 3]. For a decision to be shared there must be: an acknowledgement that there is a decision to be made; a full understanding of the risks, benefits and consequences of treatment options; and the clinician's advice and the patient's values and preferences must be considered [4]. As well as being important from medicolegal [5] and ethical perspectives, SDM can improve patient outcomes and experience as well as the communication of risk [6]. The choice of treatment is likely to be highly dependent upon individual preferences, but these preferences need to be well informed. These are complex decisions where patients and clinicians may benefit from support to enhance decision-making. We also know that application of SDM in practice remains both challenging and sub-optimal [7], hence the recent development of NICE guidelines on implementation of SDM [2]. Encouragingly, there is recent suggestion of greater effort on implementation [8].

he aim of this study was to understand SDM practice in AAA surgery and how this could be improved. The objective was to explore patient experiences of decision-making through semi-structured interviews and digital recordings of consultations.

## Methods

### Research design

This was qualitative study using in-depth semi-structured interviews. Due to the exploratory nature of the study a qualitative approach was considered more appropriate.

### Ethical approval

Ethical approval was obtained from National Research Ethics Service Committee West Midlands—South Birmingham.

## Setting

A large UK vascular unit in secondary care undertaking approximately 100 aneurysm repairs annually.

## Sample

We wished to recruit a purposive sample of patients from who had recently made a treatment decision and undergone AAA intervention (Group 1) and those at the point of making a treatment decision (Group 2). We hoped to recruit up to 12 Group 1 patients, and for Group 2 record up to 15 consultations and interview 7–10 patients shortly afterwards.

## Identification of patients and approach to participate

Clinicians identified patients who had recently undergone AAA intervention (Group 1) from electronic records. They were given study information following their post-operative clinic appointment that enabled them to contact the research team directly if they were interested in participating. Group 2 were identified at the Multi-disciplinary Team meeting and received information prior to attending the clinic to say their consultation would be audio-recorded if they agreed. On attending they were also asked if they would be happy to be contacted subsequently for interview. Given this opportunistic nature of recruitment, we are unable to determine response rates to invitations to participate.

## Data collection

Patients were offered the option of face to face or telephone interviews. Interviews were conducted by qualitative researchers experienced in both modes. It has been demonstrated that there is no difference in quality between data collected face to face and by telephone [9]. Interviews were conducted using a topic guide developed by the team, and digitally recorded. Verbal consent was obtained at the interview, for both the interview and (for Group 2) use of the consultation data. Interview and consultation sound-files were transcribed verbatim.

## Analysis

Data were analysed thematically using Framework analysis (Table 1) [10]. Two researchers developed, tested and finalised the interview data framework. A small number of transcripts were double coded as a quality assurance measure. NVivo was used as a management tool [11]. Comparisons were made between Group 1 and 2 responses to determine whether the longer period between clinic attendance and interview had an impact on recall. For the consultation data the OPTION categories were used as a coding frame without scoring [12].

**Table 1. Steps in framework analysis.**

| Familiarisation | Listen to sound files and read a small number of transcripts to identify recurrent themes |
|---|---|
| Develop thematic framework | Emergent themes from the data along with issues and questions related to the original aims and objectives of the studies brought together into a framework |
| Test and refine framework | Framework applied to a 'new' batch of interview transcripts and amended accordingly |
| Coding | Each transcript coded using the final revised framework. |
| Mapping and interpretation | Themes are mapped to look for associations and define concepts |

## Findings

Fifteen patients were interviewed, 13 by telephone. Fourteen were male and age range was 56 to 85 years. Table 2 provides information on the interviewees' age, sex, procedure (Group 1 only) and AAA size. Data were collected from Group 1 (mid to late 2017) soon after their first post-operative follow up and Group 2 (2018/19) soon after their last consultation. Two patients agreed to the recording of their consultations but were interviewed post-surgery (one felt too anxious to participate in the interview prior to his surgical procedure and the other had his procedure before an interview could be arranged). Both are included in the post-surgery group. We wanted to capture the clinical encounters in a range of clinicians to determine whether there were differences in exploring patient preferences and understanding, some key elements of SDM. Eight consultations were recorded with seven different clinicians.

The gap in data collection was due to our mode of recruitment. We had identified a junior member of the clinical team to recruit patients and co-ordinate the recording of clinical consultations. Unfortunately, this arrangement failed, and the study was put on hold until this issue was resolved. In 2018 we secured the help of two senior registrars working with the clinical lead who supported data collection.

### Diagnosis and management options

Seven interviewees attended the clinic with a family member and eight alone. The benefits of family members attending were they 'knew what was going on' (P05). Another described the impact of relaying information to family not been present at the outpatient clinic visits: *'To be honest, watching the concern on their faces was making me more concerned.' (P04).*

At the initial outpatient clinic (Table 3), most reported receiving a brief explanation of the potential management options, albeit often in a superficial way, with the caveat that further tests and fitness may impact on the options available to them. Clinicians' use of visual aids, a diagram or drawing at the time to facilitate explanations, was helpful. Clinicians sometimes added the details of the potential procedures to this diagram (Table 3).

**Table 2. Patient interviewee sex, age, surgical procedure (Group 1 only) and AAA size.**

| ID | Sex | Age | Open/EVAR | AAA size |
|---|---|---|---|---|
| **Group 1 Post surgery** | | | | |
| P01 | M | 78 | OPEN | 5.6 cm AAA |
| P02 | M | 67 | OPEN | 6.5 cm AAA |
| P03 | F | 59 | FEVAR | C5.5 cm juxtarenal AAA |
| P04 | M | 65 | OPEN | 5.8 cm AAA |
| P05 | M | 72 | OPEN | 5.5 cm AAA |
| P06 | M | 57 | OPEN | 6 cm AAA |
| P07 | M | 80 | OPEN | 5.6 cm AAA |
| P08 | M | 71 | EVAR | 7 cm AAA |
| **Group 2–making a treatment decision** | | | | |
| P09 | M | 68 | N/A | 5.6 AAA |
| P10 | M | 71 | | 5.6 AAA |
| P11 | M | 77 | | 6.1 AAA |
| P12 | M | 70 | | 5.8 AAA |
| P13 | M | 65 | | 5.4 AAA |
| P14 | M | 65 | | 6.3 AAA |
| P15 | M | 64 | | 7.5 AAA |

**Table 3. Information on AAA and its management.**

| |
|---|
| *Information from the cardiovascular team* |
| **Diagnosis and management options** |
| *'Well, I didn't know what an aneurysm was. Anyway, he explained, he gave us a quick 30 second explanation, "The veins from your heart, in your case, the lower veins from the heart have expanded, or one of them has expanded like a balloon." He explained immediately. He said, "It can burst. . . .and if it gets to that stage where it's burst, it's doubtful that you could be saved".'* **P01—INCIDENTAL—POST-SURGERY GROUP** |
| **Diagrams** |
| *'Then he explained well, until you get the scan, they'll not know because if it's twisted or whatever the one up the groin with the thing might not take hold. Okay so that's the one I wanted. He drew me a diagram on the thing and showed me all the–you know, what would be happening and everything else.'* **P14—AAA PROGRAMME—PRE-SURGERY GROUP** |
| **Treatment and risks** |
| *'Well, just that the stent was easier to do than the big one but I mean, I went through all the tests and I was fine, you know. As I'd never had surgery before I didn't know what to expect, so I just said, "Go ahead and do it."* **P02 –INCIDENTAL—POST-SURGERY GROUP** |
| *'They explained that the stents are only kind of temporary and they may have to be replaced over a period of time whereas the repair that he's doing, that should be it you know, no more aneurysm.'* **P10—AAA PROGRAMME—PRE-SURGERY GROUP** |
| *Other sources of information* |
| *I didn't know what it was until- they gave me a leaflet and all the diagrams were on it and it explained everything on the leaflet. So, I learnt everything off the leaflet. [. . .] That was really good because you saw exactly where the swelling was in your aorta, you saw. It was good that, the diagram.'* **P07 –INCIDENTAL–POST SURGERY GROUP** |
| *As soon as I told (son), he was straight on it. So maybe if you do point people in the right direction, they'll not get frightened or less frightened.'* |
| **P03 –INCIDENTAL–POST-SURGERY GROUP** |

The consultation data revealed different but comprehensive explanations of the condition. All clinicians first checked the patient's understanding of AAA. Two clinicians drew pictures to illustrate their explanations. They used images to describe the aneurysm and its risk of bursting, and analogies to explain the structure of the blood vessels and the aorta. It had previously been explained to one patient as a motorway with roads branching off, but the clinician described the AAA and system as a tree rather than continue with the analogy the patient was familiar with.

## Treatment and risks

All but two recalled receiving information about EVAR and open repair. Several remembered the finality of open repair 'once it's done, it's done', whereas a stent involved follow-up and possible further intervention. For a few their recollection was of the basics, i.e. open surgery involved a large incision, and EVAR did not. Others remembered specific pros and cons of EVAR, the shorter recovery and that it could move. One recalled three options believing open surgery's insertion of a tube (the graft) to be a different procedure. Another said at the first consultation only EVAR was discussed, and open repair introduced at a subsequent visit. Finally, one remembered discussions centred on a stent with little or nothing about open surgery.

In the recorded consultations both treatment options were described, or it was clear these had been discussed previously and the patient's level of understanding had been checked. One interviewee appeared unaware the stent would require follow up. However, the consultation data revealed the clinician had pointed out the need for annual scans following a stent.

Regarding risk, some commented the clinician conveyed the risks of *not* repairing the AAA or framed this as an option. When asked to recall the risks patients recalled the higher risk

from open surgery. Two reported the risks of both options were not conveyed by the clinician. One said the conversation centred on the intervention the clinician believed was the most appropriate. The consultation data refutes this as the clinician spent some time outlining the risks of the anaesthetic, of the two interventions and used the term 'risk(s)' 13 times. The other remembered the benefits of the interventions being discussed. The consultation data revealed the clinician mentioned risks though not in detail and spent less time discussing these (the term 'risk(s)' was used four times) and the patient responded to say they had read about the risks. As the clinician highlighted the risks and benefits together (and in that order), it is possible the latter was more likely to be retained by the patient.

**Other sources of information.** We explored whether interviewees had received or sought additional information (Table 3). One had received written information about AAA through the National AAA Screening Programme (NAAASP). Another could not remember receiving any written information, but several recalled being given a leaflet about AAA when they attended the outpatient clinic. A few commented that this, along with the verbal explanation, was helpful.

Some–and in some cases their family members—had searched online, and the information found frightened one interviewee. Another would have appreciated some direction from the clinical team on the most appropriate sites to access, '*As soon as I told (son) he was straight on (internet). So maybe if you do point people in the right direction, they'll not get frightened or less frightened*' (P03). Another who accessed a video of the procedure post-surgery was relieved they had not done so beforehand and stated, '*If I'd seen that before I'd gone into hospital, I may just have chickened out.*' (P06). Most had not looked for further information on the internet, some because they rarely used it, felt they did not require any further information, or believed it would not be informative.

## Treatment decision-making

When intervention for AAA is indicated, the patient's first decision is whether to intervene at all. Only one interviewee admitted they had considered leaving it to chance as his AAA had been slow to increase in size. Of the rest, only one other alluded to this higher-level decision, stating they felt there was little choice but to have surgical intervention as the alternative was the AAA bursting and '*you have very little chance of getting near the hospital in time*' (P07). This interviewee knew of someone who had died from an aneurysm which may have made the dangers more real.

**Influences on choice of procedure.** For two interviewees clinical factors influenced the choice of treatment, for example, only open surgery was offered to one who reported '*They couldn't put stents in, it was too serious for that*' (P06). In an unusual situation, the other interviewee stated a decision on the most appropriate procedure would be made at the time of the operation as their aneurysm was an hourglass shape. This person went into surgery unaware which procedure would be conducted.

Where both open surgery and EVAR repair were feasible the clinician recommended a particular procedure which appeared to have influenced three interviewees (Table 4). For the remaining nine, those with a preference for EVAR cited the short recovery period '*three days as opposed to 10 days*' as the reason. Additionally, longer-term monitoring with EVAR, considered beneficial, swayed another interviewee's decision. Those attracted to open surgery considered it a long-term solution; they did not want any further future risk of intervention.

Most interviewees mentioned the results of their fitness tests and were understandably happy these demonstrated they were fit; sometimes this also pleased the clinician. Although no one explicitly gave fitness as a factor that had influenced their own decision, this emphasis on

**Table 4. Treatment decision making.**

| *Influences on treatment decisions* |
| --- |
| 'Well, (clinician) is a bit of an authority on that sort of thing, so I just listened to what he had to say and agreed with whatever he and his team were proposing, which was to fit, I think they call it the stent. [...] Well, the man's an expert in his field. Apparently he gets a very good name.' **P01 –INCIDENTAL–POST-SURGERY** |
| 'I was directed by the surgeon yeah, because of the longevity of the repair yeah? So, no I didn't even consider having erm a stent [Mm] so I was quite happy yeah.' **P10 –AAA PROGRAMME–PRE-SURGERY** |
| 'I chose that one because they said once it was done it was fixed. . .. With the other one I'd still need to keep being checked and I could be back to have further procedures done. What I was thinking was, well I was 64 years old, in another 10 years' time if I'm still here I'll be 74 and would I be as fit and as able to go through a big procedure like that as I would now? So, let's just go for it.' **P04 –AAA PROGRAMME—POST SURGERY** |
| *Shared decision?* |
| 'Well I think it will be a shared decision because I would take their advice. If they said I could have both, the one I'd want is the stent because it's a shorter recovery period. However, if it's open surgery then it's open surgery so I'm not particularly concerned but I do feel that I was involved.' **P15 AAA PROGRAMME–PRE-SURGERY** |
| 'I wasn't asked which I preferred you know, I was waiting for that opportunity, but what came was you know 'You are fit, you are healthy, you are 65, you'll get through this' and that's the end of it and 'You'll get through the bigger open surgery operation and you'll be fine and that's the way to go' and (laughing) my wife was with me, we came away and went "Well that didn't quite go to plan" II don't want to say that somebody pushed you into it, I would say it was assumed that that's the way you would want to go.' **P13 –AAA PROGRAMME–POST-SURGERY** |
| 'He said I had two choices, the keyhole surgery or the other one. But, the keyhole surgery could slip and I might have to go back maybe a few times. So, he said, "I'll put you in for tests and see if you're fit enough to go through the big operation." After the tests he sent for me . . . and explained everything to me. He told me, the big operation he would cut me up the stomach and he said I was fit enough. So, I just said, "Okay." I think it was worth the risk, you know. So, that was it. I took the risk and everything couldn't have turned out better.' **P07—INCIDENTAL–POST-SURGERY** |

their level of fitness, particularly in relation to the open surgery, may have had some impact. Interestingly, one interviewee thought fitness determined the choice of intervention and assumed as he was fit, he would have the open surgery.

**Shared decision?.** The majority were happy with the decision-making process, but it was often unclear whether the treatment decision was really shared. One interviewee, who was awaiting further scans before making a final decision, believed it would be a shared decision. A few reported the clinician appearing to be more directive in the decision (interviewees were willing to accept their view) but usually giving a reason for the recommended procedure. Only one interviewee expressed disappointment in the treatment discussion (Table 4). He had accessed information, was keen on EVAR, and expected to discuss his preference but did not recall any discussion. Open surgery was recommended and he felt unable challenge the clinician in the consultation. The interviewee said, '*We came out of there and I'm thinking well okay, I've got to go through all these fitness tests and everything else, there may be time to alter this' (P13).* This account suggests the clinician based their decision on the patient's age, fitness and health rather than the interviewee's preferences.

Different clinicians sometimes proposed a different 'best' option without reference to the patient's preferences. A stent was recommended to one interviewee but subsequently another clinician proposed open surgery; the justification being that open surgery was a 'permanent fix' whereas a stent may require future intervention. He reported the clinician saying '*Think about it, you're 70 now, you'll be 80 when you have another operation. Will your fitness be the same or will it go down?" (P05).* He also recalled the clinician stating '*I'm absolutely snowed under with 'revisits'—people who are leaking and have to come in and have surgery*"'; this may have influenced the interviewee's decision to have open surgery.

The consultation data revealed one instance of the clinician exploring the patient's preferences due to lifestyle. The patient was a carer for a relative living in another part of the country.

*Clinician*: *'I think from what you told me before you're pretty active for your age, you enjoy a good retirement, you travel around and do things. . .[. . .] I really need to just make sure that we've got an agreed decision between us what we're going to do.'*

*Patient*: *'Well the last time we spoke you were on the telephone and I was happy with either but the less recovery time seems the option. We're without transport, I can't drive at the minute because of the situation we're in and the sharper it's out of the way the better it will be for us.'*

There was little evidence of this happening in the other consultation recordings. In one the clinician checked the patient was aware of the options and asked what their thoughts were. The patient responded that they had read open surgery was the best option for healthy people. The clinician then moved on to explain the risks and benefits of the two options and recommended open surgery. The remainder of the consultation focused on post-surgery issues driven by the patient.

Based on interviewees' memories of discussions leading up a treatment decision, it did appear at times that the disadvantages of EVAR was used as a justification for promoting open surgery. There was no exploration of preferences, particularly if the patient was fit: '*He said I had two choices, the keyhole surgery or the other one. But the keyhole surgery could slip, and I might have to go back maybe a few times'. (P07).*

Another interviewee's memory was of being told the disadvantages of the stent procedure. However, the consultation data reveal both options were conveyed in a balanced way and the patient said they had already discussed the two interventions with their family members and had made their decision to have open repair.

None of the interviewees mentioned using a decision support tool and none was offered in clinic, despite their availability via NHS England and NAAASP.

## What would improve information provision and decision-making?

**Provide information earlier.**   The majority expressed satisfaction with the information they received. Interviewees acknowledged the information can be alarming and the way it is conveyed and the level of detail is important. A few interviewees alluded to the amount of information provided and any more would have been too much for them personally saying '*some things are best left unsaid*' (P06).

Distributed information, where patients are given brief information initially and followed up with a more detailed explanation about AAA management, was appreciated. Earlier provision of information about the options without the pressure of having to make a decision was also helpful. Two interviewees were critical of the information provided. One had found the written information from the hospital frightening but added they were terrified generally about undergoing surgery and little would have alleviated their fear. The other felt there was a bias towards open surgery in the way the verbal information was conveyed in the consultation:

*'There's such a subtlety in words isn't there? Having the open surgery, you'll be fixed for life, that's it, walk away, never see us again. The endovascular surgery: you may need another operation, it's possible, but may and possible are very slightly different words and it depends whether you listen to the person who is talking to you or whether you read the two websites. Those words shift around a little bit. So trying to judge just how permanent or how good the endovascular surgery is becomes more of a grey area.'*

Some suggested potential areas for improvement were information on the expected recovery time, for one interviewee this had been eight rather than 'two to four weeks' (P08), and the

**Table 5. Improvements in information provision and treatment decision making.**

| |
|---|
| *'I spent a lot of time reading up on the whole thing before I went to . . . that consultation erm and I still got dumped with the six page thing (print out from the internet) that I really hadn't time to read and digest. I wonder whether other people might not do the research I did, so maybe some sort of information pack the week before the appointment . . . sent out in the post?' **P13 -AAA Programme–Pre-surgery*** |
| *'I think when you first go to screening you should be warned about what happens when it gets . . . big enough to be referred to the main hospital.. . . What they did tell me, you will be referred to the Freeman Hospital and you'll see a consultant there and he will decide the next course of action, but they didn't tell me what the course of action would be.' **P10—AAA Programme–Pre-surgery*** |
| *'Maybe if there was a group of people who'd actually had it and said what their experience was after it, that might be helpful but I'm not sure because at that stage, as I keep saying, all you're concerned with is the big picture.' **P04—AAA Programme–Post-surgery*** |
| *'Maybe the thing is the people who are doing this day in and day out you know that's their job and that's fine and but the person that they are talking to, this is a whole new experience. I find . . . if somebody says something to you that kind of goes against what you'd thought, you'll end up focused on that. The recommendation was for open surgery and suddenly you are thinking yourself 'Crumbs I read about that and it sounds like quite an operation' and you are not listening any more. Maybe the interview could be done in a slightly different way to give you time to assimilate what's been said because it's a one off thing to you, you've never had this before.' **P13—AAA Programme–Pre-surgery*** |

timing of receiving information about the procedures (Table 5). One thought the National Abdominal Aortic Aneurysm Screening Programme should provide detailed information earlier, to better prepare patients for when they reach the point where intervention is required. The other that information from the hospital, prior to the consultation, would facilitate understanding.

Another interviewee in the post-surgery group was experiencing numbness along the length of the surgical scar but thought post-operative side effects could be so varied it would be pointless to highlight this beforehand. They suggested hearing about others' experiences of the post-operative period could be useful, though prior to surgery patients' focus will be on wider issues.

**More time for decisions.** The majority reported satisfaction with their experience, even though at times it did not appear to be a shared decision. One interviewee described how they had been distracted in the consultation when the clinician proposed a procedure that was not their preferred option. They suggested clinicians may have lost touch with the fact that this is a new experience for patients who need more time to decide (Table 5).

## Discussion

This study highlighted four areas of improvement in decision-making for AAA repair. These include the timing of information and signposting high-quality information, time to consider treatment options, and the attendance of family members at clinic appointments.

The study was undertaken before the publication of NICE guidelines which have further emphasised the need to discuss options with patients, albeit with a recommendation favouring open surgery for those who are eligible [13]. Nonetheless there remains a focus on shared decision making regarding the choice between intervention and conservative management, and between open repair and EVAR in certain circumstances. This has been controversial leading to requirement for any patient choosing EVAR to be entered on the National Vascular Registry.

A major part of decision-making is having appropriate and balanced information on the treatment choices and an awareness of the risks, benefits and consequences [4]. Consistent with the literature, information from the clinical team was not always balanced with some reports of only one option being discussed [14]. In the UK there is a large regional variation in

the proportion of patients treated with EVAR (from 20% to 97%) that is unlikely to be due to patient variation [15] and may be attributable to clinician preference and experience. In our own study, there was possible bias in the risk communication of the EVAR procedure, with an emphasis on future intervention, in comparison to open surgery.

Comparison of interview and consultation data revealed patients may not recall the full discussion, or certain aspects are more prominent in their minds. Patients need sufficient time to digest information and consider their treatment preferences; ideally this process should span more than one consultation. Information needs vary between patients, and verbal information can be overwhelming particularly for those who attend without the support of another. It was unclear what written information was provided by the clinical team, as accounts varied. Regarding timing, there was a preference to receive written information earlier for men under surveillance and also prior to attending outpatients. As other research has reported [16], the information did not address the needs of all patients/family members; some sought information on-line without the guidance of the clinical team. Information about AAA available online can be difficult for patients to understand [17, 18] and may be inaccurate or not apply to their own situation.

Others have shown patients prefer SDM in vascular surgery [19]. However, as patients are not always clear about the concept of SDM, this should be explained [20]. In the current study there were few examples of clinicians exploring which treatment was appropriate considering the patients' lifestyle and home circumstances. Overall, there were low levels of engagement in the process, with patients being offered "the best" intervention. Clinicians based their own treatment decision predominantly on clinical factors rather than patient preference. Misunderstanding about treatment options suggests clinicians may not routinely or adequately check patients' understanding. This accords with earlier research where vascular surgeons rarely ask patients if they have understood the information given and their desired level of involvement in SDM [20]. From the accounts of interviewees, there was a sense that treatments were sometimes framed in a certain way, or in conjunction with other factors such as the patient's level of fitness, to justify advising a particular intervention. It is hard to determine if this influenced interviewees' treatment decisions though unbalanced information and poor risk communication can adversely affect decision-making [21, 22]. One other factor of note was how frightened some interviewees were at the thought of undergoing surgery, yet they did not feel able to raise this with the clinical team. Balanced information is also important to reduce anxiety experienced by patients and their families [16].

## Strengths and limitations

A key strength of this study is the additional knowledge it adds to the current literature on treatment decision-making in vascular surgery, particularly the comparison of interview and consultation data. One limitation is that it was based in a single vascular unit. Relying on busy clinicians and doctors on rotation meant an extended recruitment process and we were unable to achieve our original sample size of 17–20 interviews and 15 consultations. However, the interviews produced rich and useful data.

In summary, the findings highlight the vulnerability of patients in the face of complex information and a major decision. There was high praise for the clinical team though the introduction of processes to facilitate the discussions could improve the experience for patients both in terms of full information provision and their involvement in treatment decisions. A simple intervention such as including the 'Ask three questions' leaflet [23] with information/appointment letters would support patients to instigate discussions. Some guidance on trustworthy internet sites would be prudent, as it is highly likely people will search even if advised not to.

One other practical suggestion is to encourage patients to attend with a friend, carer, or family member to help patients to relay information to others and support the retention of information received. Communications with patients (directly or 'talking heads') who have undergone the procedure may provide some reassurance to patients who have concerns.

A recent umbrella review concluded that extra time and resources will have little impact on the practice of SDM, without educating clinicians on the need to build good patient relationships [24]. Training for clinicians can increase their confidence in SDM and improve risk communication [7, 25–27] and decision support tools can ensure the information provided is balanced and draw attention to the importance of patient preferences and values [28–31]. Tools have been developed specifically for SDM in vascular surgery [32]. These should also facilitate a two-way conversation and limit situations where the clinician conveys a large amount of information and suggests a treatment option without exploring patient preferences.

## Acknowledgments

We would like to thank the patients and clinicians who took the time to participate in this study.

## Author Contributions

**Conceptualization:** Gerry Stansby, Richard G. Thomson.

**Data curation:** Jan Lecouturier.

**Formal analysis:** Jan Lecouturier.

**Funding acquisition:** Gerry Stansby, Richard G. Thomson.

**Project administration:** Jan Lecouturier.

**Writing – original draft:** Jan Lecouturier.

**Writing – review & editing:** Gerry Stansby, Richard G. Thomson.

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
