## [Decision Letter · Decision Letter 0]

27 Jul 2023

PONE-D-23-01838Information provision and decision-making in the treatment of abdominal aortic aneurysm: a qualitative study of patient experiencePLOS ONE

Dear Dr. Lecouturier,

Thank you for submitting your manuscript to PLOS ONE. After careful consideration, we feel that it has merit but does not fully meet PLOS ONE’s publication criteria as it currently stands. Therefore, we invite you to submit a revised version of the manuscript that addresses the points raised during the review process.

We look forward to receiving your revised manuscript.

Kind regards,

Roberto Scendoni

Academic Editor

PLOS ONE

Journal Requirements:

Reviewers' comments:

Reviewer's Responses to Questions

**Comments to the Author**

1. Is the manuscript technically sound, and do the data support the conclusions?

Reviewer #1: Yes

Reviewer #2: Yes

2. Has the statistical analysis been performed appropriately and rigorously? 

Reviewer #1: N/A

Reviewer #2: I Don't Know

3. Have the authors made all data underlying the findings in their manuscript fully available?

Reviewer #1: No

Reviewer #2: Yes

4. Is the manuscript presented in an intelligible fashion and written in standard English?

Reviewer #1: Yes

Reviewer #2: No

5. Review Comments to the Author

Reviewer #1: This is an interesting qualitative study examining shared medical decision making (SDM) in the context of abdominal aortic aneurysm (AAA). The authors recruited 15 patients (8 post surgery, 7 making a decision regarding intervention) and conducted interviews as well as reviewed recordings of medical consultations when available for those making a decision. Several themes were identified, and suggestions were made in terms of improvement in the SDM process. Particularly compelling were the data that compared the recordings of the medical consultations with the information provided by patients during their interviews with the investigators. My specific comments follow:

Major Comments:

1. The area of SDM is important and research in this area is growing and improving. Interventions regarding AAA provide a compelling and fertile area for this research. As such, this manuscript addresses an important area of medical practice and does so in an appropriate and particularly interesting context.

2. The manuscript is generally well-written with minor exceptions (listed below). The patient accounts are absorbing and enlightening.

3. Though there is much information presented, and I found all of it engaging, I thought the primary findings/recommendations could stand out a bit more. I’m not sure the correct format (or wording) for this but I found myself at the end of the manuscript retaining a great deal of information but not quite sure how the authors would prioritize it. I don’t intend to imply that this information is absent from the manuscript, and it is quite possible that I just missed it in my reading, but I had a hard time determining the most important conclusions that the work presented both in terms of clinical application and subsequent research focus.

4. The manuscript clearly makes a contribution to the field of SDM but I am not certain of the strength of that contribution. Perhaps this is related to #3 above.

5. I think more development of the issues surrounding what an objective review of the consultation recording revealed vs. what the patients heard/remembered from it would increase the strength of the contribution of this manuscript. Clearly this is a key area of concern and one that is likely to be ground zero for miscommunications, or at least misremembering of communications, between the parties involved.

Minor Comments:

1. Throughout the document terms such as "a small number" and "several" are used. Please present the exact number.

2. There are several minor typos that should be corrected.

3. When endovascular repair is first mentioned please provide the (EVAR) abbreviation so that this is then clear in Table 2. Probably readers who study SDM in areas apart from AAA will be interested in the manuscript and may not be aware of the initialism.

4. In the research design sentence, suggest adding "a" between "was" and "qualitative" and suggest deleting "more" given that there is no referent for the implied comparison.

5. I didn't understand what was meant by the sentence in the last line of the Identification of patients section - it begins "Given this opportunistic...."

6. I was unclear what the "OPTION categories" were in the analysis section.

7. In the tables it was unclear to me what "Incidental" meant.

Reviewer #2: it is not specified after how long the patients were interviewed. It would be interesting to know because the elapsed time influences the memory.The work speaks not so much of the quality of the information provided, but of how much patients remember

6. PLOS authors have the option to publish the peer review history of their article (what does this mean?). If published, this will include your full peer review and any attached files.

Reviewer #1: No

Reviewer #2: No

---

## [Author Response · Author response to Decision Letter 0]

4 Sep 2023

Reviewer #1: This is an interesting qualitative study examining shared medical decision making (SDM) in the context of abdominal aortic aneurysm (AAA). The authors recruited 15 patients (8 post surgery, 7 making a decision regarding intervention) and conducted interviews as well as reviewed recordings of medical consultations when available for those making a decision. Several themes were identified, and suggestions were made in terms of improvement in the SDM process. Particularly compelling were the data that compared the recordings of the medical consultations with the information provided by patients during their interviews with the investigators. My specific comments follow:

Major Comments:

1. The area of SDM is important and research in this area is growing and improving. Interventions regarding AAA provide a compelling and fertile area for this research. As such, this manuscript addresses an important area of medical practice and does so in an appropriate and particularly interesting context.

2. The manuscript is generally well-written with minor exceptions (listed below). The patient accounts are absorbing and enlightening.

3. Though there is much information presented, and I found all of it engaging, I thought the primary findings/recommendations could stand out a bit more. I’m not sure the correct format (or wording) for this but I found myself at the end of the manuscript retaining a great deal of information but not quite sure how the authors would prioritize it. I don’t intend to imply that this information is absent from the manuscript, and it is quite possible that I just missed it in my reading, but I had a hard time determining the most important conclusions that the work presented both in terms of clinical application and subsequent research focus.

We thank the reviewer for their very helpful comments. We have amended the penultimate and final paragraph of the Discussion section and we hope this now more clearly states the conclusions from this research for the reader and its contribution to the field of SDM. 

4. The manuscript clearly makes a contribution to the field of SDM but I am not certain of the strength of that contribution. Perhaps this is related to #3 above.

Please see response to point 3. 

5. I think more development of the issues surrounding what an objective review of the consultation recording revealed vs. what the patients heard/remembered from it would increase the strength of the contribution of this manuscript. Clearly this is a key area of concern and one that is likely to be ground zero for miscommunications, or at least misremembering of communications, between the parties involved.

We have added more text to the Discussion to draw out and summarise the differences between the consultation and interview data.

Minor Comments:

1. Throughout the document terms such as "a small number" and "several" are used. Please present the exact number.

We understand that the use of terms such as ‘several’ can be frustrating for some readers. However, it is not customary in qualitative research to provide the exact numbers which is more in line with a quantitative paper. One disadvantage of using numbers is that they can, as Maxwell states, infer ‘greater generality for the conclusions than is justified, by slighting the specific context within which this conclusion is drawn. A particular setting or sample may be unrepresentative, and a facile reading of quantitative results may lead a reader to ignore this limitation.’ (Maxwell 2010)

*Maxwell JA. Using numbers in qualitative research. Qualitative inquiry. 2010 Jul;16(6):475-82.

2. There are several minor typos that should be corrected.

All authors have read through the manuscript and hope we have found and corrected the minor typos.

3. When endovascular repair is first mentioned please provide the (EVAR) abbreviation so that this is then clear in Table 2. Probably readers who study SDM in areas apart from AAA will be interested in the manuscript and may not be aware of the initialism.

Thank you for drawing our attention to this. We have made this change.

4. In the research design sentence, suggest adding "a" between "was" and "qualitative" and suggest deleting "more" given that there is no referent for the implied comparison.

We have made these changes.

5. I didn't understand what was meant by the sentence in the last line of the Identification of patients section - it begins "Given this opportunistic...."

Thank you for this comment. Patients in Group 2 (not yet had surgery) were identified by the clinical team as eligible after discussion by the Multi-Disciplinary Team. The clinical team did not record the numbers they approached who declined to participate. We have taken out this sentence as it refers only to Group 2 and, as the reviewer states, is confusing to the reader. 

6. I was unclear what the "OPTION categories" were in the analysis section.

OPTION – Observing patient involvement (Research Version page 89) is a 12 item instrument to measure patient involvement in consultations. We have added this to the manuscript.

7. In the tables it was unclear to me what "Incidental" meant. 

We have added a footnote to Table 3 to explain the term ‘incidental’.

Reviewer #2: it is not specified after how long the patients were interviewed. It would be interesting to know because the elapsed time influences the memory. 

Group 1 were interviewed within three months of their procedure and Group 2 within five days of their last consultation. We have added this to the manuscript.

The work speaks not so much of the quality of the information provided, but of how much patients remember. However, it is interesting to know what patients think about the information received. It is also interesting that they reported the dialogues they had with patients, but the sample is too limited. I believe that these considerations expressed should be brought within the limits of the study

We thank the reviewer for their comments. We have drawn attention to the limited sample size in the Strengths and Limitations section.

---

## [Decision Letter · Decision Letter 1]

11 Oct 2023

Information provision and decision-making in the treatment of abdominal aortic aneurysm: a qualitative study of patient experience

PONE-D-23-01838R1

Dear Dr. Lecouturier,

We’re pleased to inform you that your manuscript has been judged scientifically suitable for publication and will be formally accepted for publication once it meets all outstanding technical requirements.

Kind regards,

Roberto Scendoni

Academic Editor

PLOS ONE

Additional Editor Comments (optional):

Reviewers' comments:

Reviewer's Responses to Questions

**Comments to the Author**

1. If the authors have adequately addressed your comments raised in a previous round of review and you feel that this manuscript is now acceptable for publication, you may indicate that here to bypass the “Comments to the Author” section, enter your conflict of interest statement in the “Confidential to Editor” section, and submit your "Accept" recommendation.

Reviewer #1: All comments have been addressed

Reviewer #2: All comments have been addressed

2. Is the manuscript technically sound, and do the data support the conclusions?

Reviewer #1: Yes

Reviewer #2: Yes

3. Has the statistical analysis been performed appropriately and rigorously? 

Reviewer #1: N/A

Reviewer #2: Yes

4. Have the authors made all data underlying the findings in their manuscript fully available?

Reviewer #1: No

Reviewer #2: Yes

5. Is the manuscript presented in an intelligible fashion and written in standard English?

Reviewer #1: Yes

Reviewer #2: Yes

6. Review Comments to the Author

Reviewer #1: The authors have responded satisfactorily to nearly all of my concerns from the first draft - and those concerns were not extensive in the first place. I believe the manuscript has the potential to make a contribution to the literature in this area, not only for abdominal aortic aneurysm but in other areas of shared decision making more generally. I have a few possible minor edits to bring to the authors' attention and then one major comment on a minor point.

In the Research design it appears that the “a” between “was” and “qualitative” is still missing.

P. 8 – “…impact of relaying information to family not been present…” Should it be “…who had not been present”?

Maybe a typo in the patient report on p. 13 at top: See last sentence “…II don’t …”

In Table 5, first report, second line, is “erm” correct?

Response to the authors' reply to a minor point raised in the initial review. The authors response to my minor comment, #1, regarding providing the number of incidents, etc. rather than words like “several” or a “small number” is difficult to address with a cool mind. Off the top, it is tough to understand why authors would want to fall on a sword regarding this quite minor point and, further, would view it as an opportunity to educate the reviewer on how work is “customarily” presented in qualitative studies. So, let’s walk through it. First, the authors are, themselves, not consistent in their recommended practice. In the abstract they report “…one of the eight recorded consultations,” clearly a violation. For that matter, why did they inform us that they had 15 patients in total, 8 in one group and 7 in the other. Why not just say we had a small number of patients? In fact, we are provided quite a bit of numerical information about these patients and how they were treated in the study (e.g., 13 interviewed by phone); which is fine by me, the more information I have the better I can evaluate the research. We are told that “Two clinicians drew pictures…”, etc. There are many other examples, including within the patients’ reports (see pgs. 9, 10: Treatment and risks, p. 12 “For the remaining nine,…”). The point being that much of this exact type of information is in the report. It is, therefore, hard to see why, in certain areas, the authors draw what is apparently a firm, if somewhat arbitrary, line in the concrete. Second, the authors cite Maxwell (2010) as being concerned that if authors provide readers with too much numerical data there is the risk of readers implying “greater generality for the conclusions than is justified” and that “a particular setting or sample may be unrepresentative, and a facile reading of quantitative results may lead a reader to ignore this limitation.” Thank you for watching out of your readers and helping them improve their ability to understand and generalize from research, however, I doubt that readers need such help and if you want to provide it, there are better ways. You have a total, convenience, sample of 15, that is really two samples of 8 and 7, taken from one setting. This isn’t a sample that “may be” unrepresentative; it is almost certainly a sample that is unrepresentative. Are you really concerned that if you provide some additional data in your study this is going to lead readers to inappropriately generalize? In my initial review I did not criticize your sample, in part because with a study of this nature (and one referred to as exploratory) there is clearly no intent to generalize in the scientific use of that term. But if overgeneralization is your concern, I suggest rather than being vague in your report, you simply emphasize those limits in your limitations section or even throughout the manuscript if it is that concerning. Further, you can’t tell us what number of transcripts were double-coded, only that it was a “small number” (p. 6)? How is providing this methodological detail going to unnecessarily bias a reader’s understanding of your work except, perhaps, to be critical of exactly what that small number is? Finally, the authors seem to assume that I am unaware of qualitative research and its customs. Though it is true that most of my work has been in the quantitative vein, I have published several qualitative studies and served as the journal editor for others. Not once have reviewers criticized us for providing the actual data (e.g., 7 of 13, etc.) in the study or asked us to remove it. Further, there are many published cases where this ‘custom’ of not providing numerical data is apparently not upheld. The authors have provided a fine, interesting, and worthwhile piece of work, one that will advance the field. Thus, it remains hard to understand why they wanted to take this stand, and felt the need to be didactic, on this particular, relatively minor, issue.

Reviewer #2: The author has modified the work by integrating the requested chance. The current version of the work meets the expected requirements. It offers good scientific information that can support further studies.

7. PLOS authors have the option to publish the peer review history of their article (what does this mean?). If published, this will include your full peer review and any attached files.

Reviewer #1: No

Reviewer #2: No

---

## [Editor Report · Acceptance letter]

13 Oct 2023

PONE-D-23-01838R1 

Information provision and decision-making in the treatment of abdominal aortic aneurysm: a qualitative study of patient experience 

Dear Dr. Lecouturier:

I'm pleased to inform you that your manuscript has been deemed suitable for publication in PLOS ONE. Congratulations! Your manuscript is now with our production department. 

Kind regards, 

on behalf of

Dr. Roberto Scendoni 

Academic Editor

PLOS ONE